



# Additional soil organic carbon storage potential in global croplands

José Padarian[1], Budiman Minasny[1], Alex B. McBratney[1], and Pete Smith[2]

[1]Sydney Institute of Agriculture & School of Life and Environmental Sciences, The University of Sydney, New South Wales, Australia
[2]Institute of Biological & Environmental Sciences, University of Aberdeen, Aberdeen, UK

**Correspondence:** José Padarian (jose.padarian@sydney.edu.au)

**Abstract.** Soil organic carbon sequestration (SOCseq) is considered the most attractive carbon capture technology to partially mitigate climate change. However, there is conflicting evidence regarding the potential of SOCseq. The additional storage potential on existing global cropland is missing. SOCseq is region-specific and conditioned by management but most global estimates use fixed accumulation rates or time frames. Here, we show how the SOC storage potential and its steady state varies globally depending on climate, land use and soil. Using 83,416 soil observations, we developed a quantile regression neural network that quantifies the SOC variation within soils with similar characteristics. This allows us to identify similar areas that present higher SOC with the difference representing an additional storage potential. The estimated additional SOC storage potential of 29 to 67 Pg C in the topsoil of global croplands equates to only 2 to 5 years of emissions offsetting and 32% of agriculture's 92 Pg historical carbon debt estimate due to conversion from natural ecosystems. Since SOC is temperature-dependent, this potential is likely to reduce by 18% by 2040 due to climate change.

## 1 Introduction

Soil organic carbon (SOC) is an important component of the biosphere, controlling important physical, chemical and biological processes and their interactions. It is also a key component of soil quality and productivity (Tiessen et al., 1994), a reason why it is critical for food, soil, water, and energy security (McBratney et al., 2014). SOC oxidation is also part of the global carbon cycle by which $CO_2$ fixed by land plants can return to the atmosphere (Schlesinger and Andrews, 2000). On the global scale, agriculture is an important contributor to the increasing greenhouse gas emissions (IPCC report, August 2019) due to the elevated losses from land clearance and conventional agricultural management practices such as tillage (Russel et al., 1929; Mann, 1986; Davidson and Ackerman, 1993). The various management practices have different effects on SOC stocks (Lal et al., 2004), and the magnitude of that effect varies depending on local conditions (Blanco-Canqui and Lal, 2009; Scharlemann et al., 2014).

The "carbon debt" derived from agriculture poses the challenge of how to reverse the process. An estimate of 31.2 Pg C has been lost in topsoil (0–30 cm) through 12,000 years of agricultural land use (Sanderman et al., 2017). SOC sequestration has been considered as a viable alternative to recoup part of this debt and to partially mitigate climate change by offsetting part of the greenhouse gas emissions derived from anthropogenic activity (Minasny et al., 2017). However, the additional SOC storage potential and its controlling factors are poorly understood at the global scale. Currently, SOC storage potentials



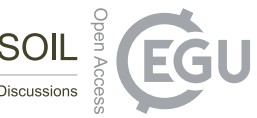

are often expressed as constant, per-year values based on management tables and assuming a uniform time to reach a new equilibrium, which is usually the 20 years default value used by IPCC (Fuss et al., 2018). Additionally, since climate exerts an essential control on SOC and global climatic data is readily available, most studies focus on temperature and precipitation to model SOC retention (Jones et al., 2005; Kramer and Chadwick, 2018; Shi et al., 2020). However, this effect of climate on SOC might not be well represented due to other confounding, unmeasured variables acting as limiting factors (Cade and Noon, 2003). Management practices are in this category, which, at the local scale, play an essential role, but detailed information with global coverage is not available to date, reason why it is usually missing in global studies.

To partially overcome the lack of detailed management data, it is possible to make some assumptions to build a predictive model. Given a set of SOC observations from locations with similar soil forming factors (which are accounted by the model), we hypothesise that differences in SOC are mainly driven by management. Different parts of the statistical distribution of SOC should represent locations with different management practices, with "good" practices in the higher quantiles and "bad" practices in the lower quantiles Fig. 1. The model is not aware of the specific management at each location but it is capable of representing different management levels and their impact on SOC given real world data. This is exactly what can be achieved using the quantile regression framework (Cade and Noon, 2003) which is what we use in this work.

The aim of this study is to estimate the additional SOC storage potential in global croplands. We focused our analysis on the top 30 cm of soil since it accounts for a large proportion of the SOC stored in soils (Batjes, 1996; Jobbágy and Jackson, 2000) and it is considered the depth that can be effectively managed to capture carbon (VandenBygaart et al., 2011) and presents faster turnover time (Shi et al., 2020). Since temperature and precipitation are important controlling factors, we also evaluate how climate change projections might affect this additional storage potential.

## 2 Methods

### 2.1 Data sources and preparation

The soil information corresponds to a compilation of different databases, including WoSIS (Batjes et al., 2020), CHLSOC (Pfeiffer et al., 2020), and the data described in (Stockmann et al., 2015). Since this a collation of mostly legacy data, and soil analytical methods change in time, part of the SOC data (Stockmann et al., 2015) was harmonised to a common measuring method using pedotransfer functions (Skjemstad et al., 2000). The observations correspond to soil profile data with SOC measurements at varying depths, which were standardised to the 0–30 depth range using the equal-area spline algorithm (Bishop et al., 1999). The soil bulk density used to calculate carbon stocks was estimated using the pedotransfer function (Adams, 1973):

$$BD = \frac{100}{\left(\frac{1.72OC}{BD_{\text{organic}}}\right) + \left(\frac{100 - 1.72OC}{BD_{\text{mineral}}}\right)} \qquad (1)$$



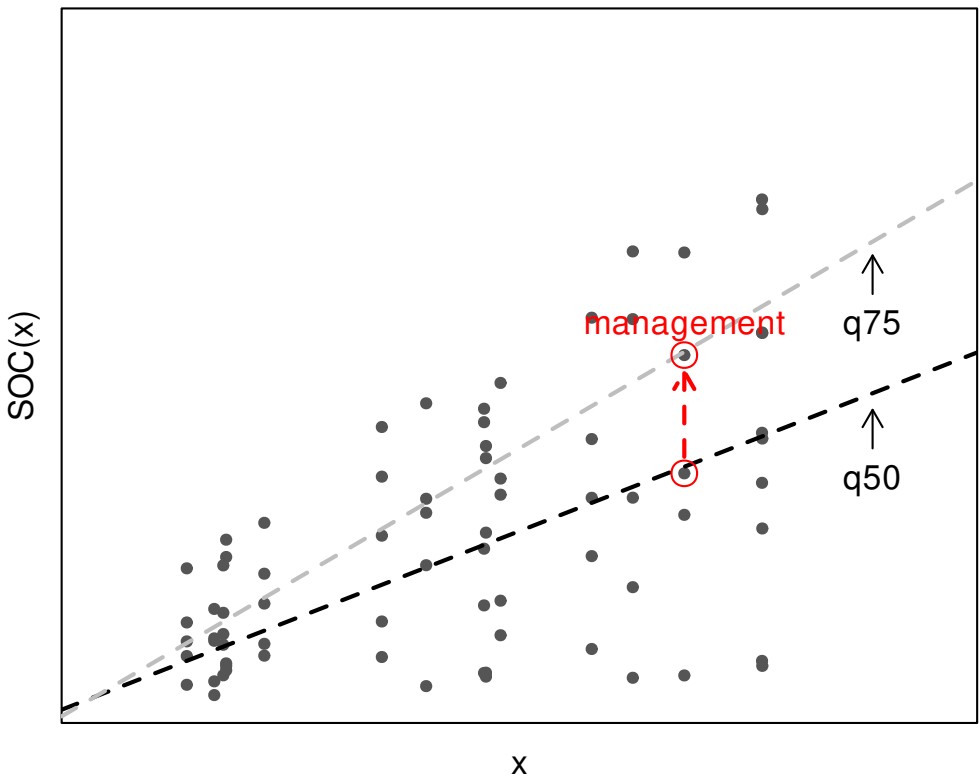

**Figure 1.** Diagram of a linear quantile regression fitted to the $50^{th}$, $75^{th}$ percentiles to explain SOC content based on a covariate $x$. The points correspond to different sites, where the lower values of SOC are due to unmeasured limiting factors (such as management). A conventional regression model is usually adjusted to the mean (close to "q50") but a quantile regression is also capable of capturing the response of high-performing sites ("q75"). Since our model includes soil, climate and topography, we hypothesise that the difference is mainly due to management practices.

where $BD_{\text{organic}}$ and $BD_{\text{mineral}}$ are the densities of the organic and mineral fractions, 0.223 and 1.32 respectively, and 1.72 is the factor used to convert from OC to organic matter content. The use of pedotransfer functions and standardisation introduces

5   uncertainties that will be fully accounted and propagated in a future study but that are unlikely to change the trends and conclusions of this study.

Land cover information was extracted from the MCD12Q1v6 MODIS product, generated by the Land Processes Distributed Active Archive Center, U.S. Department of the Interior and U.S. Geological Survey (DOI: 10.5067/MODIS/MCD12Q1.006), specifically the IGBP classification (Loveland and Belward, 1997) matching the year of each sample.

From the initial 83,416 samples, 5% was held out as a test dataset. The remaining 95% was split into training and validation datasets using a bootstrapping routine (Efron and Tibshirani, 1993) in order to find the optimal set of hyperparameters. The

covariates used as predictors include: a) digital elevation model (GTOPO30 (USGS EROS, 2020)), which is provided at 30 arc-



second resolution; and b) long term mean annual temperature (MAT) and total annual rainfall (TAP) derived from information provided by WorldClim (Hijmans et al., 2005), at 30 arc-second resolution. All data layers were resampled to a 500 m grid and standardised using the mean and standard deviation estimated from the training dataset.

## 2.2 Quantile CNN model

In this work we used a simple fully-connected, multi-task neural network with three hidden layers of 20 units each and ReLu activation functions. Since we were interested in predicting multiple sections of the SOC distribution simultaneously, the head of the network consisted of five branches of a single unit with linear activation, which corresponds to the $25^{th}$, $50^{th}$, $75^{th}$, $90^{th}$ and $95^{th}$ percentiles. Multi-task neural networks (i.e. that predict multiple targets simultaneously) have shown excellent predictive capability compared with predicting a single target in digital soil mapping and we refer the reader to Padarian et al.

(2019) for a detailed introduction. Our results are only based on the $50^{th}$, $75^{th}$ and $90^{th}$ percentiles and the $25^{th}$ and $95^{th}$ percentiles where included for regularisation (Ruder, 2017). The model was trained during 100 epochs, using a batch size of 32 samples and a learning rate of 0.001. For each percentile, the loss is estimated by:

$$\frac{1}{n}\sum_{i=1}^{n} max\left\{\tau(y_i - \hat{y}_i), (\tau - 1)(y_i - \hat{y}_i)\right\} \tag{2}$$

as per (Koenker and Hallock, 2001), where $\tau$ is the corresponding percentile and $n$ is the number of training samples. The

final, total loss corresponds to the sum of the five individual losses.

Since most global SOC models use a central estimate, such as the mean or median, here we assume the median ($50^{th}$ percentile) as the current state of SOC in the world. Higher quantiles represent situations where better management practices are in place (Fig. 1). These represent locations in the world with similar climate, soil, topography, and land use where higher SOC content values can be observed. To avoid considering very extreme cases and ensure that the target SOC content was

reached in an important proportion of the locations, we used a regression to the $90^{th}$ percentile as a technical maximum. Acknowledging that increasing SOC content is a challenging task and this technical maximum might not be always achievable, we considered a regression to the $75^{th}$ percentile as an intermediate, more achievable storage goal.

## 2.3 Model interpretation

To understand how the different covariates control SOC distribution and to corroborate that our model is capturing sound

relationships, we used an approximation of Shapley values (Lundberg and Lee, 2017) (SHAP), to estimate the contribution of each covariate to the model predictions. This is a seldom used method in soil sciences but it has been applied to large extent digital soil mapping showing large potential to interpret complex models (Padarian et al., 2020).

## 2.4 Future climate projections

To estimate the carbon stocks and additional storage capacity under future climate projections, we ran our model using down-scaled temperature and precipitation estimates of nine General Circulation Models (BCC-CSM2-MR, CNRM-CM6-1, CNRM-





ESM2-1, CanESM5, GFDL-ESM4, IPSL-CM6A-LR, MIROC-ES2L, MIROC6, and MRI-ESM2-0) for the moderately pessimistic SSP3-7.0 scenario, which considers a world that does not enact climate policies. Here we present the mean estimate of the aforementioned models for the 2021–2040 period. All the estimates were retrieved from the WorldClim 2.1 database (Fick and Hijmans, 2017).

## 3 Results and discussion

### 3.1 SOC and controlling factors

First, we considered croplands and pastures/natural ecosystems, and our model showed an upward trend from the current state to the achievable potential and technical maximum, with median SOC contents around 1.47% for the current state, 2.06 and 2.81% for the $75^{th}$ and $90^{th}$ percentile in croplands. The $75^{th}$ and $90^{th}$ percentile in pastures were higher by 2.43 and 3.43%, respectively. To illustrate the spatial variability of the additional storage potential, we generated global maps of the

SOC median, $75^{th}$ and $90^{th}$ percentile (Fig. 2). The maps show the expected global SOC trend, with low SOC contents in areas such as the Sahara, Arabian, Gobi, Australian and Atacama deserts, and high SOC contents in the humid tropics and towards the poles. The maps also show the increasing concentrations of SOC in the current, $75^{th}$ and $90^{th}$ percentiles (Fig. 2a, Fig. 2b and Fig. 2c, respectively).

The SHAP values corroborated that the model captured sensible relationships between the environmental covariates and

SOC distribution (Fig. 3a-d). Here, we describe the SOC dependence on environmental factors, but there are also intrinsic edaphic factors that control soil carbon storage. Soil clay content has been recognised as a key factor in SOC stabilisation (Oades, 1988; Schimel et al., 1994) and, ideally, our model should include this significant relationship. We used global soil texture information (Hengl et al., 2017), however, the resulting model did not show the expected spatial global patterns. This is probably due to the current global texture maps not capturing enough local variation. In consequence, we excluded clay content

from our model but we stress the need to add it to local models when good covariates are available.

For all the SOC percentiles, climatic variables have the greatest influence, closely followed by elevation and land use. All the contributions increase at higher percentiles as a consequence of the higher predicted SOC values. To normalise these values, we calculated the percentage change from the $50^{th}$ percentile which revealed a greater influence of land use (64.16 and 257.95% to the $75^{th}$ and $90^{th}$ percentiles) when compared to other environmental factors (33.09 and 107.24% for MAT; 33.07 and

60.62% for TAP; 37.29 and 54.92% for elevation). This result clearly shows the influence of land use on SOC stocks. At the median (Fig. 3a), the model assigns a modest negative contribution to land use. As we move towards higher percentiles, the negative effect of croplands (blue dots in the land use row) increases substantially, clearly differentiating itself from the other two classes (pastures and forest). The contribution of MAT also increases considerably, particularly for the observations with low temperatures, due to the decrease in carbon turnover (Carvalhais et al., 2014).

The dependence of SOC on temperature and precipitation at the global scale has been thoroughly described in the literature.

5 Our model captured this dependency, showing a clear interaction between both factors. In Fig. 3d-e, using SHAP values, we can distinguish three groups of SOC responses. First, for temperatures above 17.5 ℃, the TAP dependency follows a logistic

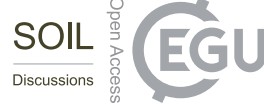

**Figure 2.** Global maps of topsoil organic carbon predictions under (**a**) current condition, (**b**) $75^{th}$ and (**c**) $90^{th}$ percentile.




curve with a relatively low plateau due to the high soil respiration rates and SOC turnover, corresponding mainly to tropical and dry regions. A second intermediate group, with temperatures between 10.0 and 17.5 ℃, also follows a logistic curve but with a higher plateau compared to the previous group. A third group, with temperatures bellow 10.0 ℃, shows an unbounded and

10    broader range of contributions within a narrower range of precipitation, describing circumpolar and alpine regions. In addition, our results also show an important threshold of around 1,000 mm of annual rainfall where the contribution of TAP becomes positive.

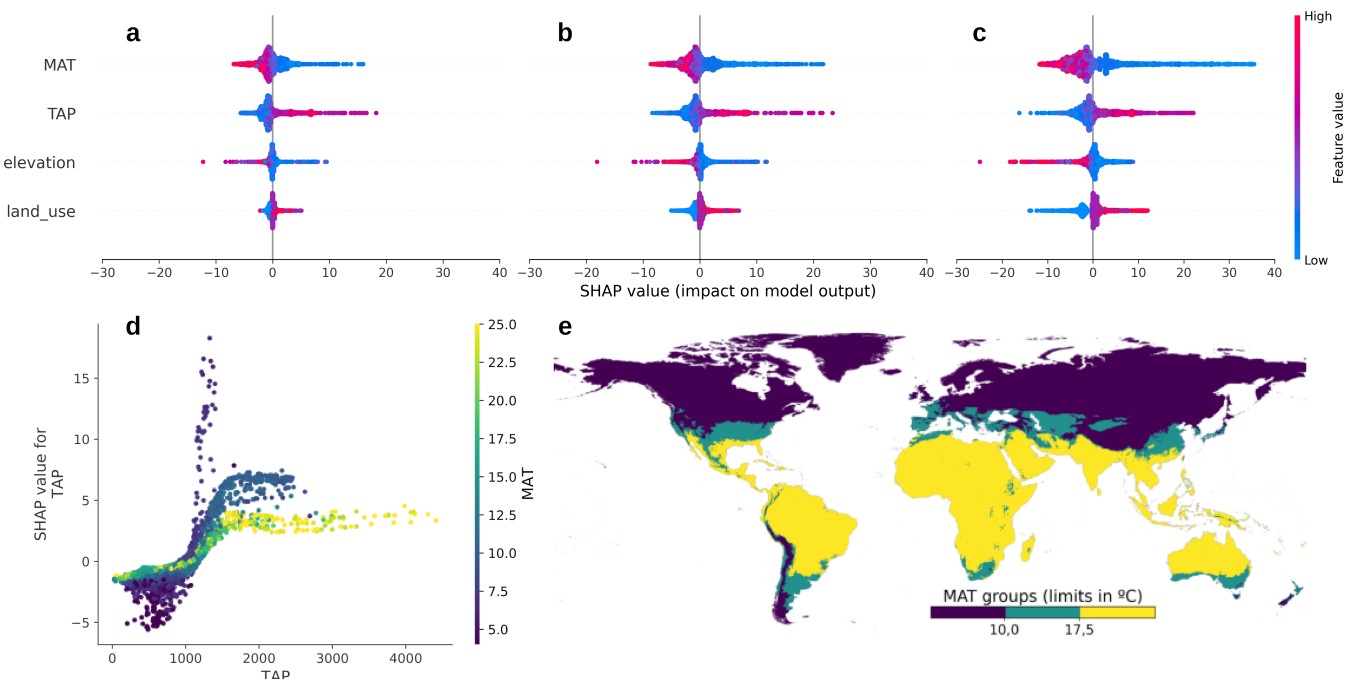

**Figure 3.** Contribution of each covariate to the final map model predictions and important interactions (SHAP values). Contributions to the $50^{th}$ (**a**), $75^{th}$ (**b**) and $90^{th}$ (**c**) percentiles. The land use covariate consists of three classes, namely croplands, pastures, and forest (red, purple and blue, respectively). **d-e** Soil organic carbon dependency on temperature and precipitation. (**d**) Global distribution of the three identified temperature groups and (**e**) precipitation dependency (SHAP values).

## 3.2    Global additional SOC storage potential

Considering the difference between a) the current and most common practices described by the central tendency of the SOC distribution ($50^{th}$ percentile) and b) the higher ends of the SOC distribution ($75^{th}$ and $90^{th}$ percentile) of similar soil under similar climate and defined land use as the additional storage potential, we estimated its magnitude and spatial distribution at the global scale. Our results show that the soils with the highest SOC additional storage potential are located towards the

circumpolar region, which corresponds to areas with high carbon density (Stockmann et al., 2015). Continental climates present



the highest SOC additional storage potential (2.81 kg m$^{-2}$), followed by Temperate and Polar/Alpine regions (both with 1.92 kg m$^{-2}$), Tropical regions (1.59 kg m$^{-2}$) and Dry regions (1.12 kg m$^{-2}$).

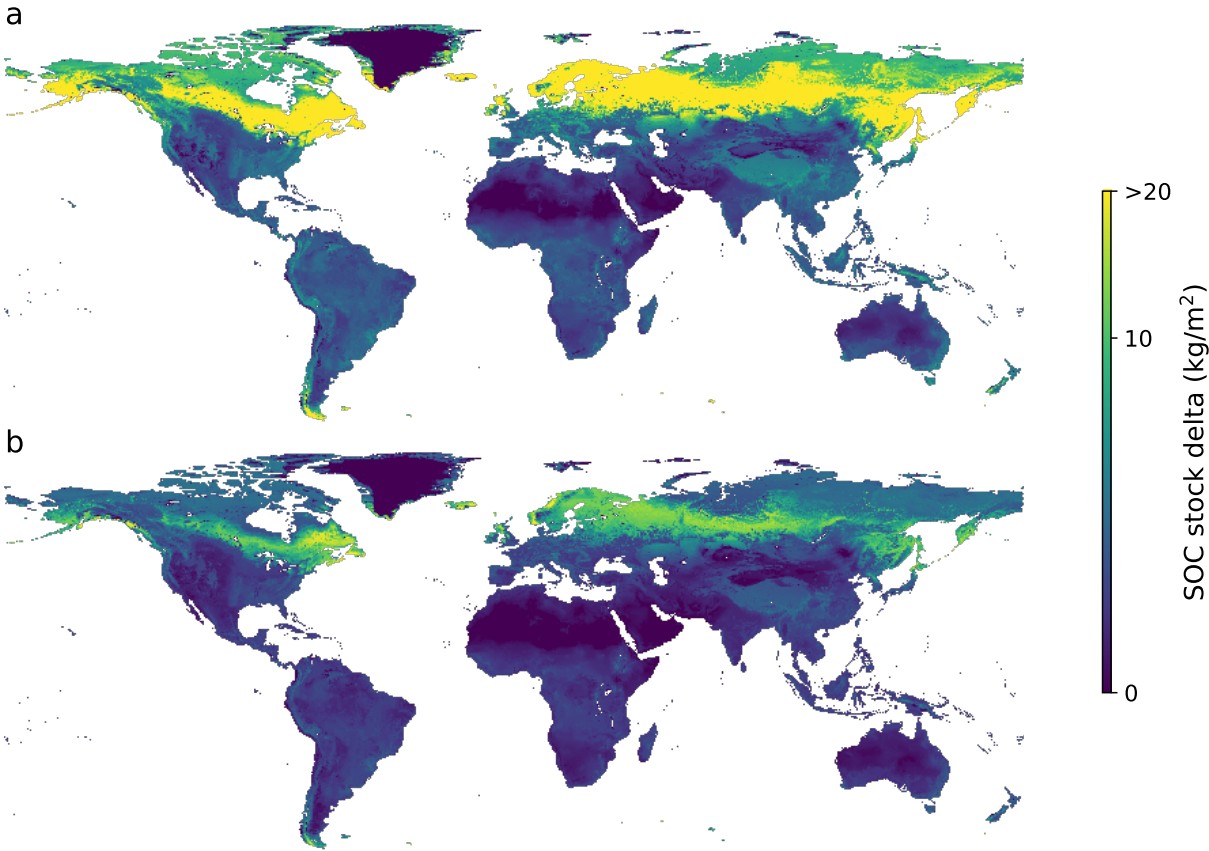

**Figure 4.** Amount of soil organic carbon required to reach the maximum soil carbon storage capacity to the technical maximum ($90^{th}$ percentile; **a**) and to the more feasible scenario ($75^{th}$ percentile; **a**).

Clearly, not all land uses can be managed for SOC sequestration, thus we focussed on the topsoil (0–30 cm) of global croplands (1,404.28 million hectares) which, according to our model ($50^{th}$ percentile), currently contain an estimate of 73 Pg C. Our results show that the areas with the greatest SOC additional storage potential are in located the high pre-cultivation carbon density ecoregions (e.g. Boreal Forests/Taiga, Flooded Grasslands and Savannas, and Temperate Broadleaf and Mixed Forests). Under the practicable scenario, the potential additional topsoil OC storage (the difference between $75^{th}$ and $50^{th}$ percentiles) of global cropland is 29.0 Pg C. Ecoregions of Temperate Broadleaf and Mixed Forests, and Temperate Grasslands, Savannas and Shrublands account for 60.4% of the total additional storage potential with 17.5 Pg C. For the technical maximum (difference between $90^{th}$ and $50^{th}$ percentiles), the total potential is 66.6 Pg C, and both ecoregions account for a similar proportion (60.9%).





Recent years have seen increased interest in the potential of improve SOC stock in croplands. For instance, the "4 per mille" initiative, launched during COP21 in December 2015 (Minasny et al., 2017), estimates that increasing SOC stocks by $4\text{‰ yr}^{-1}$ could offset some fraction of annual $CO_2$ emissions into the atmosphere. There has been a debate on its actual potential as the assumption was made that all soils of the world would increase its SOC more or less uniformly. Accumulation

of SOC, regardless of the rate, can only be achieved for a limited time as soils have a natural upper limit for carbon storage which is also limited by management. Using our additional storage potential estimates, we generated global maps simulating a $4\text{‰ yr}^{-1}$ accumulation rate from the current condition (Fig. 5) and calculated the number of years to reach the practicable and technical maximum. It is important to note that soils will accumulate carbon at different rates, but we used the fixed rate of $4\text{‰ yr}^{-1}$ because it is currently being used to design policies in many places. Our results showed a large spatial variation

of the maximum amount of years under the 4 per mille initiative, with a median period of 94 and 216 years to reach the $75^{th}$ and $90^{th}$ percentiles, respectively. For both percentiles, the average capture rate is around 0.31 Pg C yr$^{-1}$ which corresponds to only 3.5% of the C emissions used to estimate the 4‰rate (8.9 Pg C yr$^{-1}$) (Minasny et al., 2017).

In addition to using a fixed accumulation rate, the results presented in Fig. 5 assume a linear accumulation. Of course, soils behave differently with SOC accumulation diminishing approximately exponentially in time (Minasny et al., 2017; Franzlueb-

bers et al., 2012) but it might still be a valid estimate since SOC accumulation rates, in many cases, could be greater than 4‰(Francaviglia et al., 2019). Our estimates are in line with other studies that report croplands reaching a new, higher SOC concentration equilibrium after over a century (Smith et al., 1996; Soussana et al., 2004).Regardless of the accumulation rate, the total additional carbon storage potential of the topsoil in croplands is limited. Our total estimates of 29.0 and 66.6 Pg C for the $75^{th}$ and $90^{th}$ percentiles are equivalent only to 2 and 5 years of global emissions (49 Pg $CO_2$ eq. yr$^{-1}$ of greenhouse gas

derived from anthropogenic activity (Pachauri et al., 2014)).

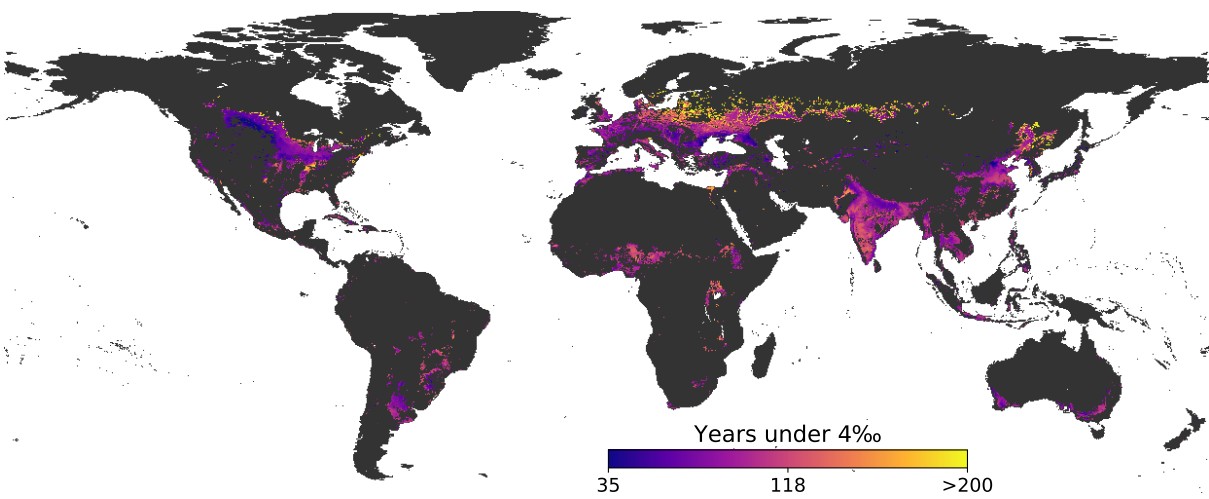

**Figure 5.** Years to reach the $75^{th}$ percentile for global croplands. This assumes a fixed $4\text{‰ yr}^{-1}$ accumulation rate which might not be adequate in all cases but it is currently used as an aspirational target.





Our practicable potential (29 Pg C) is close to the 31.2 Pg C historical debt due to agriculture estimated by a recent study (Sanderman et al., 2017), although their estimate is at the lower end of the 21–186.0 Pg C range derived from a) their 62 Pg C estimate for the current SOC stock in croplands and b) the fact that soils of agroecosystems contain 25% to 75% less SOC than their counterparts under natural ecosystems (Lal et al., 2018). Using our model, we estimated a historical carbon debt ranging between 10 and 174 Pg C by replacing all croplands with a range of region-specific natural ecosystems, from non-forest with average ($50^{th}$ percentile) carbon density to forest with high carbon density ($90^{th}$ percentile). If we consider a midpoint within the latter range (92 Pg C), our practicable potentials account for only 32% of the historical carbon debt due to agriculture (72% for the technical maximum). We only considered croplands in our analysis as these areas have lost more SOC. There is potential for managed grasslands, however currently we cannot differentiate managed and natural grassland using satellite imagery at the global level.

Compared with previous estimates, our results show a slightly higher additional carbon storage potential for global croplands. A total of 18 to 37 Pg C, under medium and high storage scenarios with accumulation rates of 0.9 and 1.85 Pg C $yr^{-1}$ and the assumption of reaching a new equilibrium after 20 years as been reported (Zomer et al., 2017), with estimates based on gridded predictions (Hengl et al., 2017) and considering a uniform sequestration rate for all croplands. A slightly wider range of sequestration potential has been reported (Lal et al., 2018) with a total of 7.63 to 43.25 Pg C over a period of 25 to 50 years, assuming the adoption of region-specific best management practices. According to an extensive review (Fuss et al., 2018), the best estimate of realistic technical potential (close to the median of the minimums of their reviewed studies) is between 20.1 and 46.2 Pg C until 2050. From that year onwards, the accumulation rates could not be maintained due to sink saturation. It is important to remember that our approach estimates the additional storage potential based on real observations, within a similar climatic context, and not on technical accumulation rates of specific management practices. Since our approach is not based on fixed technical accumulation rates, our results are not necessarily constrained to the 20–50 years period which most studies consider, and could be another reason for our higher estimates.

Several studies have raised concerns about the barriers to sequester SOC (Rumpel et al., 2020). One of the main advantages of our study is that we use a large global database and the estimates are based on real world observations, meaning that a group of locations already reached the target SOC stocks for a given combination of environmental conditions. However, a current limitation to have in mind is that our model is based on biophysical factors but does not take into account socio-economic barrier, disproportionally affecting developing countries, that can imped the adoption of new management practices.

### 3.3 Effect of climate change

An important point to consider is that the sequestration potential could vary under the future climate. Given the high dependence of SOC on temperature, it is expected that relatively fast global warming will shift most ecosystems toward a lower SOC equilibrium (Fig. 3d-e) and that this effect will be more pronounced in areas with larger SOC concentrations (Fig. 3a-c). These projections have been reported in many studies (Robinson, 2007; Crowther et al., 2016; Melillo et al., 2017) and they are likely to result in reduced sequestration potential.





Utilising CMIP6 downscaled future climate projections for a moderate "business as usual" shared socio-economic pathway (SSP3-7.0), we estimated a mean reduction of 18% in the total sequestration potential in croplands in the next 20 years, from 29.0 Pg C to 23.8 Pg C, and from 66.6 Pg C to 54.7 Pg C for the $75^{th}$ and $90^{th}$ percentiles, respectively. That estimate does not include the drop in carbon concentration of the current state ($50^{th}$ percentile) which implies an additional loss of 7.4 Pg C.

### 3.4 SOC sequestration still a priority

We have shown that total amount of additional carbon croplands can store is relatively modest compared to the sustained emission of greenhouse gases derived from anthropogenic activity. It is unreasonable to expect that a single sector can offset global emissions, especially considering their increasing trend. Nevertheless, incorporating carbon into soils by improving management practice should still be a priority to ensure food security. According to our estimates, agriculture generates a carbon debt, so we need to properly manage croplands to be sustainable and avoid the expansion of agricultural land due to loss of soil productivity.

Agricultural productivity has been directly related to SOC contents. If reduced below some critical limits, soil condition declines and so does crop yield. By increasing SOC to its practicable and technical upper limits ($75^{th}$ and $90^{th}$ percentiles), between 224 and 418 million hectares could be taken above the critical SOC limits of 1.1% and 2% for tropical (Aune and Lal, 1997) and temperate (Loveland and Webb, 2003) areas, respectively.

There are still many knowledge gaps that need to be filled and that could help to improve our current model. As mentioned in previous sections, detailed texture information at the global scale is still required and that is also applicable to other soil properties that highly correlate with SOC (Rasmussen et al., 2018). Additionally, our approach places management practices into different quantiles of a distribution based on their SOC density, but it is not capable of identifying them. More research is needed to identify region-specific management practices that can enhance soil carbon. Many countries have a national registry of lands (e.g. Land Parcel Identification in Europe (Leo and Lemoine, 2001)), at least for the management of agricultural subsidies, which should be integrated into soil information systems and added to this type of model.

### 4 Conclusions

Our results put in perspective the expectation around soil carbon sequestration in global croplands. They indicate that, even if the additional amount of carbon croplands can accumulate is large (22–66 Pg C), the process is slow, spanning periods up to over a century. Regardless of the accumulation rate, the total additional storage is relatively small compared to the sustained emission of greenhouse gases derived from anthropogenic activity, equating to only 2 to 5 years of emissions offsetting.

Regarding the historical impact of agriculture, our results suggest that the current management practices close to our $75^{th}$ percentile can only recoup 32% of our estimated 92 Pg C historical debt. Even considering the best current management practices (equivalent to our $90^{th}$ percentile), we would not be able to fully recoup that debt, only offsetting 72% of them. Hence, agriculture has an intrinsic environmental cost that needs to be taken into account for territorial planning.

Soil carbon capturing measures should be complementary to general carbon emission reduction plans in order to tackle climate change. Moreover, given the high contribution of climate to SOC accumulation, it is expected that the additional SOC storage potential will be affected by climate change if this is not the case. Our results showed that the potential could be reduced by 18% under a moderate "business as usual" shared socio-economic pathway (SSP3-7.0).

10     The total amount of additional carbon that global croplands can store is relatively small in the context of global carbon emissions. However it is critical in the context of soil and food security. We estimate that it is possible to restore between 224 and 418 million hectares if we implement management practices conducive to out $75^{th}$ and $90^{th}$ percentiles. Soil carbon is directly related to agricultural productivity and any improvements in soil condition are a step closer to more sustainable and resilient agricultural systems.

15   *Data availability.* The data used in this study is not publicly available due to third party restrictions on parts of it. Data are however available from the authors upon reasonable request and with permission of those third parties.

*Competing interests.* The authors declare that they have no conflict of interest.

*Acknowledgements.* This project was funded by the ARC Discovery project DP200102542 "Forecasting soil conditions"



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
