# Peer review of "Additional soil organic carbon storage potential in global croplands"

_SOIL, 2021_

## Referee Comment (RC1)

Philippe C. Baveye

The manuscript "Additional soil organic carbon storage potential in global croplands", submitted by Padarian et al. for publication in *Soils,* gives me from the onset two very conflicting impressions. The first is that it is a sad reflection of the dismal state of our discipline that the authors attempt to publish this manuscript in 2021 when the calculations it presents, had they been carried out many years ago as they should, would have helped avoid lots of time wasted in futile discussions about a chimera. The second feeling I have is that, after substantial revisions, this text could provide a welcome closure of sorts to a particularly dismaying episode in the history of soil science, arguably rivaling in embarrassment for our discipline with a time in the late 60s when blind opportunism led some soil scientists to be dramatically on the wrong side of history (in the "polywater" episode). I will try to explain first where these opposite feelings originate, before I provide specific suggestions to revise the manuscript in order to emphasize the message that in my opinion it should contain.

During the first decade of this century, the scholarly literature on soil organic matter dealt even-handedly with a healthy diversity of topics. Many researchers were concerned about the fact that, as a result of climate change, rising temperatures in different parts of the globe could hasten the mineralization of soil organic matter, release large amounts of $CO_2$ in the atmosphere, and thereby speed up climate change even more, to the point where it would get entirely out of control (Powlson, 2005; Baveye, 2007). The effect of rising temperatures in that context remained poorly understood, and the general feeling was that it was important to elucidate it urgently, in part to determine if it was possible practically to slow down the decrease in soil organic matter content, which, as the monitoring results of Bellamy et al. (2005) revealed, had been going on for decades. At about the same time, research on the "priming effect" indicated that any addition of fresh organic matter to soils to compensate for its faster mineralization would have to be carried out extremely carefully, because in the process one ran the risk of mineralizing "stable" or "recalcitrant" organic matter that had been residing in soils for extremely long periods of time (e.g., Fontaine et al., 2007). Progress was also achieved concomitantly on the nature and fine-scale heterogeneity of soil organic matter, using advanced spectroscopic techniques (e.g., Schumacher et al., 2005; Kinyangi et al., 2006), and on accounting for the effect of the microscale heterogeneity of the soil pore space on the dynamics of SOM mineralization by microorganisms (e.g., Kuka et al., 2007). In many ways, this period can be characterized as one where researchers had a clear perception of the utter complexity of the system they were dealing with, realized that all of its multiple facets needed to be investigated carefully, were not afraid to engage in a "science as subtle as its objects of study" (Dorit, 2011), and were making significant progress.

Then hype hit the proverbial fan. Several authors tried in their writing, and in meetings with policy makers in Brussels and Washington, to promote the sequestration of carbon in soils as a win-win scenario to mitigate climate change (e.g., Lal, 2004, 2010). At about the same time, others started promoting biochar, with vastly overblown, still unfulfilled, promises that the addition of large amounts of pyrolysed

organic matter to soils could be a "win-win-win" solution to the climate crisis. The reception of these silver bullets by decision-makers was lukewarm, at best, but that did not deter some from trying to push their agenda further behind the scene. The result of these efforts was that, at the COP 21 meeting in Paris, the then minister of agriculture of France announced that by increasing the carbon content of soils by 0.4% or 4 per 1000 per year, one could compensate for the yearly anthropic release of $CO_2$ into the atmosphere. Even though several reports published earlier by French scientists (e.g., Arrouays et al., 2002; Chenu et al., 2014) had calculated that the amount of carbon that could be sequestered overall in soils was much lower than that put forth in the 4 per 1000 proposal, the latter was nevertheless adopted by COP 21 participants and was integrated in the final reports of the conference. Criticisms of the proposal were very quick to emerge from various quarters, e.g., by White (2016) who concluded his analysis of the 4 per 1000 initiative by stating unequivocally that "sequestering carbon in agricultural soils will not provide a major offset for greenhouse gas emissions". Essentially the same opinion was widely circulated in private conversations or e-mails, for example in an e-mail I received from a prominent soil researcher, who wrote about soil carbon sequestration that he was not willing to "promise that it will deliver a complete offset of fossil C emissions, which is plainly absurd".

At that stage, given the serious doubts expressed by various people about the 4 per 1000 initiative, it would seem that the reasonable thing to do would have been to pause, proceed to further estimations (like those now described by Padarian et al. in their manuscript) of the actual carbon sequestration potential of soils, and reflect carefully about the exact potential of this approach to mitigate climate change, before any more efforts were made to convince politicians that this was the way to go. A well-known historical precedent should have guided soil researchers on how to deal properly with controversial claims made by politicians supposedly on the basis of scientific or technological ideas. In 1983, U.S. president Ronald Reagan announced the concept of the Strategic Defense Initiative (SDI), quickly nicknamed derisively the "Star Wars Program". Spearheaded by several physicists, in particular Edward Teller (the "father of the hydrogen bomb"), the initiative was nevertheless greeted by intense skepticism and even mocked as ludicrous by numerous other physicists, from the moment it was launched. After the initiative failed to deliver preliminary proofs of concept, the U.S. government asked the American Physical Society (APS) to conduct an in-depth analysis of the feasibility of the initiative and of the extent to which it was advisable to devote significant amounts of money to it. Even though physics research stood to gain a lot financially from SDI, which made some physicists reluctant to "throw away the baby with the bath water", the APS report (Bloembergen et al., 1987) nevertheless, very candidly, concluded that the technologies being considered were decades away from being ready for use, and that at least another decade of research was required to know whether what had been proposed in the SDI was even possible. Thereafter, in short order, the budget of the program was repeatedly cut, its focus severely scaled-down, and eventually the initiative was cancelled. By then, one could argue that SDI had been a political success, in that it probably helped end the cold war. However, had the APS not effectively managed to terminate it, SDI would undoubtedly have turned out to be an embarrassment of major proportions for the physics community in the U.S., because of its blatant inability to deliver on

unwarranted promises made, and because of suspicions of manipulation and fraud that inevitably would have arisen (Broad, 1992).

Regrettably, this sound precedent was not emulated in our discipline. In April 2017, Minasny et al. published an article that to casual readers (as politicians would likely be), may seem like an endorsement of the 4 per 1000 initiative by a large group of scientists worldwide. Indeed, in the abstract, the authors state that "reported soil C sequestration rates globally show that under best management practices, 4 per mille or even higher sequestration rates can be accomplished". Anyone not reading the article further (and in particular not reaching a contradictory statement a few lines down in the abstract) would likely conclude that the 4 per 1000 idea was what Lal liked to refer to as a "low-hanging fruit" that is easy to reach. The undeniable risk associated with such a perspective is that it would give ammunitions to those who want to slow down the transition to renewable forms of energy, away from the greenhouse-gas-producing consumption of fossil fuels.

If the goal of our work is to publish articles that get heavily cited, then the article by Minasny et al. (2017) has been a resounding success, since according to Google Scholar it has been cited more than 900 times so far, since its publication. However, from a more elevated perspective on science, the article was an unmitigated disaster. Contrary to the rich, multifaceted approach to soil organic matter that had prevailed a decade earlier, the various contributors to the article of Minasny et al. (2017) adopted an extremely narrow view on the topic. The effects of rising temperatures or priming on the carbon content of soils were entirely ignored, as was the necessarily linked fate of nitrogen or dynamics of microorganisms. A significant part of the earlier literature on all these issues, even though the knowledge it contained was eminently relevant, was completely bypassed. Several groups of researchers promptly wrote letters to the editor to criticize the conclusions of the Minasny et al. article (van Groeningen et al., 2017; Amundson and Biardeau, 2018; Baveye et al., 2018; Poulton et al., 2018; VandenBygaart et al., 2018; White et al., 2018). Like the APS report did a few decades ago for the SDI, the very sound criticisms raised by these various authors should have put an end to the propaganda campaign surrounding the "4 per 1000 initiative" and in particular its portrayal to decision-makers as a low-hanging fruit. But it did not, sadly, and hundreds of articles are now being published every year where researchers are trying to demonstrate that, locally under very specific conditions, the 4 per 1000 goal can be reached. Some articles go as far in their biased attempts to justify the initiative as to ignore the severe disequilibrium caused by rising temperatures or enhanced erosion in their modeling of future trends in soil carbon contents (e.g., Chenu et al., 2019)! Efforts such as these are not useful and do not warrant publication, since the fact that, under very special conditions, soil carbon content can be drastically increased was never in doubt. The Dogons in Mali demonstrated that conclusively centuries ago… What has been in doubt from the very beginning was the ability of soils to offer a solution to climate change at a large enough scale and over a long enough timeframe that it should be considered seriously by policy makers.

Amidst the mass of articles that have unfortunately been devoted to the 4 per 1000 initiative since 2016, some have carried out the kind of calculations that should have logically been carried out before the propaganda-style promotion of soil carbon sequestration began. A good example in this respect is afforded by the very comprehensive analysis carried out recently by Riggers et al. (2021) in the context of German croplands, to determine the extent to which changing climate in decades

ahead could affect soil organic carbon stocks. These authors considered 3 different climate change scenarios between 2014 and 2099, as well as a scenario assuming no future climate change. They used 5 distinct methods to estimate organic carbon inputs based on crop yields and crop-specific parameters, and adopted a multi-model ensemble consisting of five different SOC models to predict the level of organic carbon input required to reach specific SOC stocks in soils at the end of the 21st century. Their simulation results suggest, among other things, that organic carbon input to the soil in 2099 needs to be between 51 and 93% higher than what it is today just to maintain SOC stock levels at their current value. Riggers et al. (2021) eventually conclude that "under climate change increasing SOC stocks is considerabl[y] challenging since projected SOC losses have to be compensated first before SOC built up is possible. This would require unrealistically high OC input increases with drastic changes in agricultural management." In other words, their conclusions are not very different than what White (2016) had written.

Given the recent publication of this excellent article by Riggers et al. (2021), which incidentally Padarian et al. do not cite, and of other recent work that reaches the same conclusion, it seems legitimate to ask whether the manuscript by Padarian et al. comes up with anything sufficiently new that its publication is warranted. In terms of its contents, the answer to that question is negative, in my view. Enough authors have confirmed by now Arrouays et al.'s (2002) or White's (2016) assessment that soil carbon sequestration can only be a very marginal solution to climate change. In that context, Padarian et al.'s calculations add nothing. The only purpose that one more publication on the topic would achieve is to increase the number of citations of those who wrote on it earlier. That is not a valid achievement from my perspective. Nevertheless, their paper could still have some value if it were revised in such a way as to come out as an acknowledgement by the authors that the Minasny et al. (2017) article was a blunder of epic proportions, which oriented much of the research on soil organic matter in the last 5 years in a direction it should never have taken, leads nowhere, and should now be forgotten. That, hopefully, would put an end to the nonsense started in 2015 and would restore some of the credibility that our discipline has lost in the process. As some of us (Baveye et al., 2020; Vogel et al., 2021) have pointed out recently, that does not mean that research on soil organic matter dynamics, and in particular on its effect on the resilience of the architecture of soils under fast changing environmental conditions, is not needed. It is in fact direly and urgently needed for other reasons than those advocated in the 4 per 1000 chimera, and that point has to be made to policy makers, without repeating the mistakes made in the last few years, i.e., without making promises that we are not sure we can keep, or even worse, that we already know we cannot keep. Soils, and the structure and fate of soil organic matter in particular, are very complex, and we cannot promise we can come up easily with straightforward answers to the daunting questions society is asking us. But we can promise that we shall try to answer them as best we can, in direct continuation of the very nice research that was carried out a decade ago.

**Specific comments:**
Page 1, title: I find this title potentially very misleading. Some readers may derive the impression from it that, relative to what many researchers have described in recent years as the (limited) potential of soils to sequester carbon, Padarian et al. have

somehow found "additional" storage potential. That is not the case, since the conclusion of their text really only confirms the many previous assessments that have been published and that all point to the very marginal contribution soil carbon storage or sequestration might make to climate change mitigation.

Line 1 of abstract: "Soil organic carbon sequestration (SOCseq) is considered the most attractive carbon capture technology to partially mitigate climate change." The very first sentence of the article is seriously misleading as well. Certainly there are a few articles presenting SOCseq as an ideal technology to capture carbon, but there is no consensus on the matter at all. There is a huge literature reviewing in detail modern carbon capture technologies, in which authors do not mention soils at all (see, e.g., Rubin et al., 2012; Wilberforce et al., 2021). Clearly, when people outside the soil science community think about carbon capture to mitigate climate change, they overwhelmingly do not think about soils among the top 10 best candidate technologies.

Page 1, lines 24-25. Again, it is not really clear what the authors are referring to with this "additional" SOC storage potential. What is it "additional" to?

Page 2, lines 17-18. In support of their assumption that the 0-30 cm depth of the soil is where most of the SOC storage occurs, the authors cite 2 classic but relatively old references on a topic about which a lot has been written during the last 2 decades, in particular by researchers who have recommended that measurements of soil carbon storage should routinely extend deeper than just the top 30 cm. It seems to me highly desirable that the authors justify their assumption in light of this more recent literature, and not just older articles.

Page 3. Figure 1. I was taught by statistics professors never to calculate linear regression lines, especially when data points are severely scattered as is the case in Figure 1, without considering the uncertainty that is associated with the regression coefficients. I do not believe there is any reason to envisage quantile regressions differently. Also, the case is not made particularly well in the text as to why the authors chose to consider the median rather than the mean. Finally, the last sentence of the figure caption requires more information in order to be understandable. How reasonable is it to adopt the hypothesis that "the difference is mainly due to management practices"? Could other factors than soil, climate, topography, and land use account for the difference, and to what extent?

Page 4, line 21. Again, why not consider the mean? That is not clear at all. Does the choice of the median make a difference in terms of the final conclusions?

Page 4, line 30. Most readers are probably not going to be familiar with Shapley values, which have not been used much in soil science, so a more detailed introduction to them and to their advantages is indicated here.

Page 5, lines 5-6. There is a huge amount of material missing here. The authors mention that they ran all kinds of simulations with 9 general circulation models, "for the moderately pessimistic SSP3-7.0 scenario, which considers a world that does not enact climate policies". Why that particular shared socio-economic pathway scenario, among all the possible ones??? How does that choice affect the conclusions reached at the end of the article? Without a whole more information in that respect, it is hard to evaluate the simulations that have been carried out.

Page 5, lines 21-22, "Soil clay content has been recognized as a key factor in SOC stabilization". Again the authors are basing their statement on relatively old references. In recent years, various researchers have shown that in practice it is not

the clay content per se that matters, but the ratio SOC/clay content (e.g., Johannes et al., 2017; Prout et al., 2020).

Page 8, line 10, "are in located". I have a hard time understanding how, with two native English speakers as co-authors, the manuscript was submitted with obvious typos, misspellings, and syntax problems. Did they not read the manuscript, or do they simply expect someone to do the editing for them?

Page 9, lines 16 and 17, "which corresponds to only 3.5% of the C emissions used to estimate the 4 ‰ rate". Either I do not understand what the authors mean by that statement, or I do not understand how they could just mention it in passing without emphasizing how this statement challenges everything that has been claimed about the 4 per 1000 idea…

Page 9, lines 21-23, "Our estimates are in line…". In support of the statement in this sentence, the authors cite relatively old references again, but fail to point out that Franzluebers et al. (2012), whom they cite 4 lines earlier, reach a different conclusion. These authors observe that 10 years after conversion of an arable cropping system into perennial grassland — admittedly one of the fastest agricultural practices to sequester carbon in soil — the rate of C accumulation down to a depth of 20 cm drops by half, and after 20 years, it is only 0.2 Mg ha$^{-1}$ y$^{-1}$, i.e., a quarter of its initial value of 0.8 Mg ha$^{-1}$ y$^{-1}$ (see Fig. 2 in Baveye et al., 2018). After 50 years, the rate is virtually zero, and a new soil equilibrium is reached. So, at least some people have found timeframes that are much shorter than those found by Padarian et al. This point needs to be discussed.

Page 9, line 23, "total estimates". It is clear what the authors refer to with this expression.

Page 10, line 32, "our practicable potentials account for only 32% of the historical carbon debt due to agriculture". Again, this statement needs to be emphasized more than it currently is.

Page 10, line 9, "From that year onwards, the accumulation rate could not be maintained due to sink saturation". There is something missing in the narrative between the previous page and this one. On page 9, the authors are referring to accumulation timeframes of over a century, and now mention sink saturation occurring in 2050.

Page 10, line 18, "imped". English!

Page 11, line 7, "in the next 20 years". Why only 20 years?

Page 11, section 3.4. This section contains a lot of hand waving to try to justify asking for more funding to carry out research on soil organic matter, but I doubt that the arguments presented would convince very many decision-makers.

Page 12, lines 10-11, "The total amount of additional carbon that global croplands can store is relatively small in the context of global carbon emissions". It took 12 pages to get to the point where the authors concede that their conclusion is similar to what other people have said consistently since 2015, and, actually, Arrouays et al. (2002) already wrote in 2002. Hence the question I raised earlier of whether this article really needs to be published, since it contributes very little, if anything at all to the debate. As I wrote earlier, this manuscript would be useful if it were revised in a way that it carry the message that it is time to stop the nonsense, and to agree once and for all that the sequestration or even the storage of carbon in soils is nowhere near large enough to be more than a very marginal contribution to the mitigation of climate change. If the authors stated that clearly, this article might be useful to close a

parenthesis that should never have been opened, and to encourage policy-makers to focus back on societal changes that can have a real effect on climate change, such as a switch to renewal forms of energy, or a move to an economy that involves less long-distance transport of goods than is the case at the moment.

**References**

Amundson, R., Biardeau, L. (2018). Opinion: Soil carbon sequestration is an elusive climate mitigation tool. Proceedings of the National Academy of Sciences 115, 11652–11656.

Arrouays, D., Balesdent, J., Germon, J. C., Jayet, P. A., Soussana, J. F., & Stengel, P. (2002). Increasing carbon stocks in French agricultural soils? Synthesis of an assessment report by the French Institute for Agricultural Research on request of the French ministry for ecology and sustainable development. Scientific Assessment Unit for Expertise, INRA, Paris.

Baveye, P.C. (2007). Soils and runaway global warming: Terra incognita. Journal of Soil and Water Conservation, 62, 6, 139A-143A.

Baveye, P.C., Berthelin, J., Tessier, D., Lemaire, G., 2018. The "4 per 1000" initiative: A credibility issue for the soil science community? Geoderma 309, 118–123. Doi: 10.1016/j.geoderma.2017.05.005.

Baveye, P.C., Schnee, L.S., Boivin, P., Laba, M., Radulovich, R. (2020) Soil organic matter research and climate change: Merely re-storing carbon versus restoring soil functions. Front. Environ. Sci., 8, 579904. Doi: 10.3389/fenvs.2020.579904

Bellamy, P.H., Loveland, P.J., Bradley, R.I., Lark, R.M., Kirk, G.J.D., 2005. Carbon loss from all soils across England and Wales 1978-2003. Nature 437, 245-248.

Bloembergen, N.; Patel, C. K. N.; Avizonis, P.; Clem, R. G.; Hertzberg, A.; Johnson, T. H.; Marshall, T.; Miller, R. B.; Morrow, W. E.; Salpeter, E. E.; Sessler, A. M.; Sullivan, J. D.; Wyant, J. C.; Yariv, A.; Zare, R. N.; Glass, A. J.; Hebel, L. C.; APS Council Review Committee; Pake, G. E.; May, M. M.; Panofsky, W. K.; Schawlow, A. L.; Townes, C. H.; York, H. (July 1, 1987). "Report to The American Physical Society of the study group on science and technology of directed energy weapons". Reviews of Modern Physics. 59 (3): S1-S201. Doi: 10.1103/RevModPhys.59.S1

Broad, William J. (1992). Teller's war: The top-secret story behind the star wars deception. New York: Simon & Schuster.

Chenu, C., Klumpp, K., Bispo, A., Angers, D., Colnenne, C., Metay, A. (2014). Stocker du carbone dans les sols agricoles: Évaluation de leviers d'action pour la France. Innovations Agronomiques 37, 23–37.

Chenu C, Angers DA, Barre P, Derrien D, Arrouays D, Balesdent J (2019) Increasing organic stocks in agricultural soils: Knowledge gaps and potential innovations. Soil Till Res 188:41–52. https://doi.org/10.1016/j.still.2018.04.011

Dorit, R. (2011). The Humpty-Dumpty problem. American Scientist 99(4), 293-295. Doi: 10.1511/2011.91.293.

Fontaine, S., Barot, S., Barre, P., Bdioui, N., Mary, B., and Rumpel, C. (2007). Stability of organic carbon in deep soil layers controlled by fresh carbon supply. Nature 450, 277–281. doi: 10.1038/nature06275

Johannes, A., Matter, A., Schulin, R., Weisskop, P., Baveye, P., Boivin, P. (2017). Optimal organic carbon values for soil structure quality of arable soils. Does clay content matter? Geoderma, 302, 14–21. https://doi.org/10.1016/j.geoderma.2017. 04.021

Kinyangi, J., Solomon, D., Liang, B., Lerotic, M., Wirick, S., and Lehmann, J. (2006). Nanoscale biogeocomplexity of the organomineral assemblage in soil: Application of STXM microscopy and C 1s-NEXAFS spectroscopy. Soil Sci. Soc. Am. J. 70, 1708–1718. doi: 10.2136/sssaj2005.0351

Kuka, K., Franko, U., and Ruhlmann, J. (2007). Modelling the impact of pore space distribution on carbon turnover. Ecol. Modell. 208, 295–306. doi: 10.1016/j.ecolmodel.2007.06.002

Lal, R. (2004). Soil carbon sequestration to mitigate climate change. Geoderma 123, 1-22.

Lal, R. (2010) Beyond Copenhagen: mitigating climate change and achieving food security through soil carbon sequestration. Food Security. 2:169-177. Doi: 10.1007/s12571-010-0060-9

Minasny, B., Malone, B.P., McBratney, A.B., Angers, D.A., Arrouays, D., Chambers, A., Chaplot, V., Chen, Z.-S., Cheng, K., Das, B.S., Field, D.J., Gimona, A., Hedley, C.B., Hong, S.Y., Mandal, B., Marchant, B.P., Martin, M., McConkey, B.G., Mulder, V.L., O'Rourke, S., Richer-de-Forges, A.C., Odeh, I., Padarian, J., Paustian, K., Pan, G., Poggio, L., Savin, I., Stolbovoy, V., Stockmann, U., Sulaeman, Y., Tsui, C.-C., Vågen, T.- G., van Wesemael, B., Winowiecki, L., 2017. Soil carbon 4 per mille. Geoderma 292, 59–86. Doi: 10.1016/j.geoderma.2017.01.002.

Poulton, P., Johnston, J., Macdonald, A., White, R., Powlson, D., 2018. Major limitations to achieving "4 per 1000" increases in soil organic carbon stock in temperate regions: Evidence from long-term experiments at Rothamsted Research, United Kingdom. Glob. Chang. Biol. 24, 2563–2584. Doi: 10.1111/gcb.14066.

Powlson, D., 2005. Will soil amplify climate change? Nature 433, 204–205.

Prout, J. M., Shepherd, K. D., McGrath, S. P., Kirk, G. J. D., and Haefele, S. M. (2020) What is a good level of soil organic matter? An index based on organic carbon to clay ratio, Eur. J. Soil Sci., 1–11. Doi: 10.1111/ejss.13012.

Riggers, C., Poeplau, C., Don, A., Frühauf, C., Dechow, R. (2021). How much carbon input is required to preserve or increase projected soil organic carbon stocks in German croplands under climate change?. Plant Soil 460, 417–433 (2021). Doi: 10.1007/s11104-020-04806-8

Rubin, E.S., Mantripragada, A., Versteeg, P., Kitchin, J. (2012). The outlook for improved carbon capture technology. Prog. Energy Combust. Sci. 38, 630–671.

Schumacher M, Christl I, Scheinost AC, Jacobsen C, Kretzschmar R (2005) Chemical heterogeneity of organic soil colloids investigated by scanning transmission X-ray microscopy and C-1s NEXAFS microspectroscopy. Environmental Science and Technology, 39, 9094-9100. Doi: 10.1021/es050099f

van Groenigen, J.W., van Kessel, C., Hungate, B.A., Oenema, O., Powlson, D.S., van Groenigen, K.J., 2017. Sequestering soil organic carbon: a nitrogen dilemma. Environ. Sci. Technol. 51, 4738–4739. Doi: 10.1021/acs.est.7b01427.

VandenBygaart, A.J., 2018. Comments on soil carbon 4 per mille by Minasny et al. 2017. Geoderma 309, 113–114. https://doi.org/10.1016/j.geoderma.2017.05.024.

Vogel, H.-J., Balseiro-Romero, M., Kravchenko, A., Otten, W., Pot, V., Schlüter, S., Weller, U., & Baveye, P. C. (2021). A holistic perspective on soil architecture is needed as a key to soil functions. European Journal of Soil Science, 1– 14. https://doi.org/10.1111/ejss.13152

White, R.E. 2016. Myths about carbon storage in soil. Australasian Science 37(5), 37. [Available at: https://www.researchgate.net/publication/303487443_Myths_about_carbon_storage_in_soil, last retrieved October 4, 2021]

White, R.E., Davidson, B., Lam, S.K., Chen, D., 2018. A critique of the paper 'soil carbon 4 per mille' by Minasny et al. (2017). Geoderma 309, 115–117. Doi: 10.1016/j.geoderma.2017.05.025.

Wilberforce, T., Olabi, A.G., Sayed, E.T., Elsaid, K., Abdelkarrem, M.A. (2021). Progress in carbon capture technologies. Science of the Total Environment 761, 143203.

---

## Author Comment (AC1)

**Response to Philippe C. Baveye**

The vast majority of the review is a polemic about soil carbon sequestration and the 4 per 1000 concept. That discussion was held earlier and readers can look at that. Here are the relevant publications:

- Minasny, B., Arrouays, D., McBratney, A.B., Angers, D.A., Chambers, A., Chaplot, V., Chen, Z.S., Cheng, K., Das, B.S., Field, D.J. and Gimona, A., 2018. Rejoinder to Comments on Minasny et al., 2017 Soil carbon 4 per mille Geoderma 292, 59–86. Geoderma, 309, pp.124-129.

- Rumpel, C., Amiraslani, F., Chenu, C., Cardenas, M.G., Kaonga, M., Koutika, L.S., Ladha, J., Madari, B., Shirato, Y., Smith, P. and Soudi, B., 2020. The 4p1000 initiative: Opportunities, limitations and challenges for implementing soil organic carbon sequestration as a sustainable development strategy. Ambio, 49(1), pp.350-360.

- Rumpel, C., Amiraslani, F., Chenu, C., Cardenas, M.G., Kaonga, M., Koutika, L.S., Ladha, J., Madari, B., Shirato, Y., Smith, P. and Soudi, B., 2020. Response to "The "4p1000" initiative: A new name should be adopted" by Baveye and White (2019). Ambio, 49(1), pp.363-364.

In addition to that, a conference on the 4 per 1000 topic was held in June 2019 (https://symposium.inrae.fr/4p1000/) where scientist were given the opportunity to present their cases (in favour or against). The reviewer had presented his case and a paper that summarises their consensus was published:

- Amelung, W., Bossio, D., de Vries, W., Kögel-Knabner, I., Lehmann, J., Amundson, R., ... & Chabbi, A. (2020). Towards a global-scale soil climate mitigation strategy. Nature communications, 11(1), 1-10.

With respect to the current paper, our replies to the comments are as follows.

**Comments on the scope of this paper and its novelty**

Here we present a global modelling study based on a large database of soil observations (83,416 samples). We use the quantile regression framework to determine different levels of soil organic carbon (SOC) taking into account soil, climate, topography, and land use variability. Given those levels, we estimate the additional SOC storage potential in global croplands, which is clearly stated as our aim in P1L17. The combinations of dataset, scale and methodology makes this a unique study. The first study that looks at carbon potential based on global empirical data.

You mention that other publications, such as the case study by Riggers et al. (2021) in Germany, based on a simulation, reached similar conclusions to this study. Somehow you see that as something negative. The opposite view is that it is nice to see that the conclusions hold at different scales, using different datasets and methods.

**Comments on the use of the term "additional carbon storage"**

How to call what we calculated was the most challenging part. We are not referring to the maximum amount of SOC a soil can hold (associated with mineralogy). In addition to that, we are not specifically talking about a net $CO_2$ removal from the atmosphere which is closer to the definition of carbon sequestration. We use the term additional as in "in addition to the current stocks". We will expand the last paragraph of the introduction to clarify this.

That said, we are open to suggestions if the title is not clear enough. Perhaps " soil organic carbon storage potential in global croplands".

**Specific comments**

**"I find this title potentially very misleading. Some readers may derive the impression from it that, relative to what many researchers have described in recent years as the (limited) potential of soils to sequester carbon, Padarian et al. have somehow found"additional" storage potential. That is not the case, since the conclusion of their text really only confirms the many previous assessments that have been published and that all point to the very marginal contribution soil carbon storage or sequestration might make to climate change mitigation."**

As mentioned above, we will add a clarification to what we mean by "additional" and we are also open to modify the current title.

**"Line 1 of abstract: 'Soil organic carbon sequestration (SOCseq) is considered the most attractive carbon capture technology to partially mitigate climate change.' The very first sentence of the article is seriously misleading as well..."**

Changed to "Soil organic carbon sequestration (SOCseq) is considered **an** attractive carbon capture technology..."

**"Page 2, lines 17-18. In support of their assumption that the 0-30 cm depth of the soil is where most of the SOC storage occurs, the authors cite 2 classic but relatively old references on a topic about which a lot has been written during the last 2 decades, in particular by researchers who have recommended that measurements of soil carbon storage should routinely extend deeper than just the top 30 cm. It seems to me highly desirable that the authors justify their assumption in light of this more recent literature, and not just older articles."**

We wrote: "We focused our analysis on the top 30 cm of soil since it **accounts for a large proportion** of the SOC stored in soils.". That is very different to "the 0-30cm depth range is where most of the SOC storage occurs". Proportionally speaking (in terms of depth units) SOC density is higher in the 0-30 cm range since SOC content usually decreases in depth. We will add "carbon density" to be extra explicit about this.

We also gave 2 other reasons to focus on the 0-30cm range (it is considered the depth that can be effectively managed to capture carbon, and presents faster turnover times). We agree that measuring SOC at greater depths is important but perhaps more relevant in other systems such as grassland, which are not included here, as mentioned in P10L8.

Another data related reason is that many soil surveys only collect topsoil samples. Since we want to maximise the number of observations for modelling, only using observations that measured carbon to 100+cm was not an option.

**"Figure 1. I was taught by statistics professors never to calculate linear regression lines, especially when data points are severely scattered as is the case in Figure 1, without considering the uncertainty that is associated with the regression coefficients. I do not believe there is any reason to envisage quantile regressions differently. Also, the case is not made particularly well in the text as to why the authors chose to consider the median rather than the mean. Finally, the last sentence of the figure caption requires more information in order to be understandable. How reasonable is it to adopt the hypothesis that"the difference is mainly due to management practices"? Could other factors than soil, climate, topography, and land use account for the difference, and to what extent?"**

Figure one is just an explanatory diagram, not derived from our data.

We use the median to more accurately describe the central tendency since the data is positively skewed. We will update the text accordingly.

Regarding the hypothesis that "the difference is mainly due to management practices" (after taking into account soil, climate, topography, and land use variation) we think is quite reasonable. Of course, the whole process is quite complex and a model will always be underspecified, particularly at the global scale where data availability is a limiting factor. We will add some references and update the text to reflect that.

**Page 4, line 21. Again, why not consider the mean? That is not clear at all. Does the choice of the median make a difference in terms of the final conclusions?**

We will update the text to explain why we used the median. The final estimates will vary but the final trend and conclusions will be the same (a relatively small amount of extra SOC stored vs the high levels of emissions).

**Page 4, line 30. Most readers are probably not going to be familiar with Shapley values, which have not been used much in soil science, so a more detailed introduction to them and to their advantages is indicated here.**

We will add more details.

**Page 5, lines 5-6. There is a huge amount of material missing here. The authors mention that they ran all kinds of simulations with 9 general circulation models, "for the moderately pessimistic SSP3-7.0 scenario, which considers a world that does not enact climate policies". Why that particular shared socio-economic pathway scenario, among all the possible ones??? How does that choice affect the conclusions reached at the end of the article? Without a whole more information in that respect, it is hard to evaluate the simulations that have been carried out.**

We used the mean prediction of those nine General Circulation Models (i.e. a single "simulation"). We will rewrite the section to make that clear.

Regarding why the SSP3-7.0 scenario, we think it is very illustrative to show what happen if we do not do something (hence the description of a "world that does not enact climate policies"). Of course, it would be very interesting to test all possible combinations of General Circulation Models and socio-economic pathways but that is out of the scope of this paper. We will expand the section to give more details on why we picked that scenario and add a more description of it.

The conclusions would probably co-vary with the amount of change. For the moderately pessimistic scenario, we estimated that the potential would decrease by 18%. Worse scenarios will show a larger decrease and vise versa. Of course, a complete study running all the possible combinations would be necessary to know for sure, which, as we already mentioned, it is out of the scope of this paper.

**Page 5, lines 21-22, "Soil clay content has been recognized as a key factor in SOC stabilization". Again the authors are basing their statement on relatively old references. In recent years, various researchers have shown that in practice it is not the clay content per se that matters, but the ratio SOC/clay content (e.g., Johannes et al., 2017; Prout et al., 2020)**

That SOC/clay ratio does depend on clay content. We will add those reference and update that sentence.

**Page 9, lines 16 and 17, "which corresponds to only 3.5% of the C emissions used to estimate the 4 ‰ rate". Either I do not understand what the authors mean by that statement, or I do not understand how they could just mention it in passing without emphasizing how this statement challenges everything that has been claimed about the 4 per 1000 idea. . .**

4 per 1000 uses an annual emission of 8.9 Pg C. Given our estimates of potential stocks and a $4‰ yr^{-1}$ accumulation rate, that means that we could only offset 3.5% of the total C emissions ($8.9$ Pg $yr^{-1}$). We will update that sentence to clarify and emphasise how small 3.5% is.

**Page 9, lines 21-23, "Our estimates are in line. . . ". In support of the statement in this sentence, the authors cite relatively old references again, but fail to point out that Franzluebers et al. (2012), whom they cite 4 lines earlier, reach a different conclusion. These authors observe that 10 years after conversion of an arable cropping system into perennial grassland — admittedly one of the fastest agricultural practices to sequester carbon in soil — the rate of C accumulation down to a depth of 20 cm drops by half, and after 20 years, it is only 0.2 Mg ha-1 y-1, i.e., a quarter of its initial value of 0.8 Mg ha-1 y-1 (see Fig. 2 in Baveye et al., 2018). After 50 years, the rate is virtually zero, and a new soil equilibrium is reached. So, at least some people have found timeframes that are much shorter than those found by Padarian et al. This point needs to be discussed.**

In this study we are not considering grasslands but we agree that some people have found much shorter timeframes. In fact, the map under that paragraph (Fig. 5) shows a large variation in the timeframes, with a lower boundary even shorter than 50 years. We start that paragraph saying "In addition to using a fixed accumulation rate, the results presented in Fig. 5 assume a linear accumulation. Of course, soils behave differently with SOC accumulation diminishing approximately exponentially in time". Right after the sentence you mention, we add "Regardless of the accumulation rate, the total additional carbon storage potential of the topsoil in croplands is limited.".

We think that is in line with your specific comment. We will add a mention the lower limit (in addition of the upper limit of "over a century") to make that clearer.

**Page 10, line 32, "our practicable potentials account for only 32% of the historical carbon debt due to agriculture". Again, this statement needs to be emphasized more than it currently is.**

In the conclusions we write "Regarding the historical impact of agriculture, our results suggest that the current management practices close to our $75^{th}$ percentile can only recoup 32% of our estimated 92 Pg C historical debt. Even considering the best current management practices (equivalent to our $90^{th}$ percentile), we would not be able to fully recoup that debt, only offsetting 72% of them. Hence, agriculture has an intrinsic environmental cost that needs to be taken into account for territorial planning.".

We will expand that paragraph to emphasise it further.

**Page 10, line 9, "From that year onwards, the accumulation rate could not be maintained due to sink saturation". There is something missing in the narrative between the previous page and this**

**one. On page 9, the authors are referring to accumulation timeframes of over a century, and now mention sink saturation occurring in 2050.**

That comes from the review by Fuss et al. (2018), not our results. We will rewrite that sentence to make it clearer.

**Page 11, line 7, "in the next 20 years". Why only 20 years?**

That comes from the dataset used (WorldClim) which has averages over 20-year periods (2021-2040, 241-2060, 2061-2080, 2081-2100). We only used 2021-2040 since a) it is illustrative enough and b) uncertainty is lower (the uncertainty increases as we predict further into the future). We will expand Section 2.4 to clarify that.

**Page 11, section 3.4. This section contains a lot of hand waving to try to justify asking for more funding to carry out research on soil organic matter, but I doubt that the arguments presented would convince very many decision-makers.**

As you mention in page 4 of your review: "As some of us have pointed out recently, that does not mean that research on soil organic matter dynamics, and in particular on its effect on the resilience of the architecture of soils under fast changing environmental conditions, is not needed."

Research in organic matter is still needed. We point out some research topics that are directly related to this paper (that would improve the presented model) but there are many others, including the ones related to your research.

**Page 12, lines 10-11, "The total amount of additional carbon that global croplands can store is relatively small in the context of global carbon emissions". It took 12 pages to get to the point where the authors concede that their conclusion is similar to what other people have said consistently since 2015, and, actually, Arrouays et al. (2002) already wrote in 2002. Hence the question I raised earlier of whether this article really needs to be published, since it contributes very little, if anything at all to the debate. As I wrote earlier, this manuscript would be useful if it were revised in a way that it carry the message that it is time to stop the nonsense, and to agree once and for all that the sequestration or even the storage of carbon in soils is nowhere near large enough to be more than a very marginal contribution to the mitigation of climate change. If the authors stated that clearly, this article might be useful to close a parenthesis that should never have been opened, and to encourage policy-makers to focus back on societal changes that can have a real effect on climate change, such as a switch to renewal forms of energy, or a move to an economy that involves less long- distance transport of goods than is the case at the moment.**

We mention that in the abstract (page 1) but we will add that same sentence to make it more explicit. We present that in page 12 because it is a conclusion derived from the results of the analysis performed in this study.

Regarding the novelty of this paper, refer to or general comment on page 1 of this response.

---

## Author Comment (AC2)

We thank the reviewer for her/his comments. We will include your suggestions in the revised version of the manuscript.

**The quantile approach relies on a strong assumption that SOC will be same under similar soil forming factors. If SOC values are different, it much be induced by management. But the problem is that we would never find two soils with the same forming factors. Numerous factors (e.g., climate seasonality) regulate SOC dynamics and thus SOC stock at a typical site. At the same site, SOC would also experience temporal changes. In this study, only very few potential predictor variables of SOC were considered. Other variables such as soil parent material, land use history, climate variability are not included.**

This is a regression model that uses global information to represent soil forming factors in a continuous scale. Of course, given that we are dealing with real numbers, it is very unlikely that two locations have *exactly* the same values for all the factors but a regression model does not rely on that.

The process related to carbon sequestration, stabilisation, etc, is quite complex and a global model will always be underspecified. Our model is purposely simple for a simple reason. By just adding a few extra covariates, the model becomes "ill-defined" since we do not have enough data to cover the covariate space. For instance, land use history is very important but most samples sites have a single observation in time.

That said, the covariates used do explain a large part of global SOC variation. We will expand the "Data sources and preparation" section and add some references to clarify that.

**More importantly, the approach adopts another strong assumption of steady state. If the soil is not at the steady state, the approach will be invalid, because a mediate soil (50% quantile) may experience SOC loss. The SOC would be an overestimation of the real 50% quantile, and vise versa.**

This model is static, so the reviewer is right. We do assume a more or less "steady state" for croplands, which would be the assumption of any static model. However, this is probably a good approximation for two reasons. First, given the nature of our soil database, which comes from surveys not purposely designed to monitor SOC changes in time, soil samples targeting agricultural land are more likely located in well established farms. Second, part of the process of generating the landcover information used in this work (MODIS) has a spatio-temporal smoothing, which ensures that a landcover is more or less persistent (at least a few years).

Probably the worse case would be some samples taken after only 2-3 years under farming. However, the larger proportion of the carbon loss happens during that period, so the problem would be attenuated.

We are happy to expand the discussion to raise this important point.

**The manuscript paid little attention to the potential uncertainties in the relevant estimations. Two major uncertainties I think should be explicitly tested are: the approach used to estimate BD and prediction uncertainty by the quantile CNN model. To my knowledge, BD has been reported for some soil profiles, why was BD estimated using a pedotransfer function? Could you please test the credibility of the BD estimates which are vital for estimating SOC stock?**

In general, most pedotransfer function (PTF) are not universally applicable and the uncertainties vary in space. We think that an uncertainty analysis with that level of detail is beyond the scope of this paper. However, we agree with the reviewer that a general sense of the error induced by the PTF is useful. We do have BD records for part of the database and we could use to get a general estimate of the uncertainty. We will add that to the revised version of the manuscript.

It is important to note that this addition will not change the conclusion (that croplands have a limited capacity to offset carbon emissions and that the capacity changes with climate change predictions).

**In page 3, a bootstrapping routine was mentioned. The reader cannot find anything descriptions on the purpose of this routine. To predict SOC stock?**

We did that to generate multiple SOC stock maps (a map for each bootstrap iteration) and estimate a mean prediction. We will clarify that in the revised version. In addition, we will add a deviation estimate (standard deviation) to the reported mean values.

**The author very briefly described future climate projections. As the SOC estimates were conducted at the global scale, I believe historical climate records are required to run the GCMs. How were the**

**climate projections used in their models for predicting SOC stocks?**

We did not run the GCMs. As mentioned in Section 2.4, we used estimates from the WorldClim 2.1 database. We will expand that section to clarify that.

---

## Author Comment (AC3)

We thank the reviewer for her/his comments. Our response to your general and specific comments bellow.

**Carbon sequestration in agroecosystems appears to be a significant way to offset some anthropogenic CO2 emissions, and no-till is generally considered an efficient and essential component for sequestering SOC. However, data comparing no-till and full tillage show large uncertainties, and not all studies found that SOC levels increased following a change in management to no-till. While there may be a significant change in C distribution in the soil profile, this does not necessarily translate into an increase in total SOC. Since the most important management factor appears to have a limited impact, the hypothesis of this study is generally in question. So the main question is what management practices are we talking about that would result in significant SOC storage. I am assuming that what we are seeing are the effects of potential natural land cover, not the effects of human land use.**

This paper is not about the application of no-tillage to achieve additional carbon in the soils. No-tillage is only *one of the many* management practices available, which is also implemented in combination with other managements. One of the challenging things about any farm/soil management is that there is no perfect recipe that can be prescribed. Their effectiveness varies according to soil/environmental factors affecting each location, reason why spatial models like this one are so important. That is why the title of the first reference that you provide is "Climate and Soil characteristics Determine Where no-till Management can Store carbon in Soils and Mitigate Greenhouse Gas emissions". We will add that reference to our discussion.

**In this regard, land use history is also a very important factor. This is probably the most difficult part of the equation. It is likely to have a greater influence compared to changes in analytical methods over time. A common problem with global studies and modeling is spatial resolution. Land use and its history often vary on very fine scales, which cannot be accounted for with low resolution spatial data.**

We agree that land use history is very important and satellite imagery can detect that to different levels of details. Ideally we would like information at very detailed spatio-tempotal resolution to be able to detect large changes (e.g. forest $\rightarrow$ agriculture) but also smaller changes such as crop rotation. As far as we know, we are not there yet.

The product that we use (MODIS) has a spatial resolution of 250m and it *can* differentiate croplands from other categories. Of course, it could be missing borders between classes or small plots of land. There is extensive literature describing the use of MODIS products and we are happy to add a few references in the methods section.

**One factor controlling SOC distribution is soil erosion. Countermeasures may well cause SOC to accumulate in the soil. Colluvial soil can also store a lot of SOC. Estimates put the resulting global storage at 78 Pg C. Such effects are not considered in this study because neither terrain characteristics, soil properties, nor parent material are accounted for in the models. That said, the results of the SHAP analysis become clearer at the 75th and 90th percentiles. This may indeed indicate some effect of management practices, but also the general potential to develop higher SOC levels in some terrain positions, as evidenced by the increase in importance of low elevations. Again, this may be an effect of small-scale (lower elevation) terrain and soil variability - rather than management practices.**

Just for reference, the 78 Pg C you mention was estimated in a period between 6000 BC and 2015 AD (8015 years). This study is focused on a much shorter time scale. However, we acknowledge that erosion is a problem in agricultural production. For that reason, information about relief is important in SOC models. In this case, that effect is mitigated by the fact that the samples are distributed mostly in a narrow range of slope values ("flat areas") and part of it is already captured by the elevation covariate.

Regarding the SHAP values, there is a slight increase in the contribution of low elevations for Q75 but rather small compared with the large effect of temperature and precipitation.

**All analyses and results are relatively worthless if they are not validated. And here, no validation of the hypothesis and no validation statistics for the modeling are presented. Therefore, the result is relatively meaningless.**

As per response to RC2, We will add validation statistics and deviation from the percentiles derived from our bootstrapping routine.

This model includes well recognised factors that affect SOC. We acknowledge in the discussion that our model is not

capable of identifying management practices since for that we would required very detailed management information at the global scale and enough soil information to cover all the possible combinations. Unfortunately, such level of information is not available to date, to the best of our knowledge. We also mention that in the discussion.

**If I understand correctly, the global predictions are made based on cropland/pasture data only. The calculated SOC totals seem to be based on the global models. Here, however, at least the current forest areas would have to be removed, because otherwise the global storage capacity would be overestimated.**

This is indeed a global model but forest areas are not included in any estimates. If this is related to figure 2 and 4, we do not mask the maps so it easier to see the global trends. We will add that comment to the corresponding captions.